# Improved Guarantees for Offline Stochastic Matching via New Ordered Contention Resolution Schemes

**Brian Brubach**
Computer Science Department
Wellesley College
Wellesley, MA 02481
bb100@wellesley.edu

**Nathaniel Grammel**
Department of Computer Science
University of Maryland
College Park, MD 20742
ngrammel@umd.edu

**Will Ma**
Graduate School of Business
Columbia University
New York, NY 10027
wm2428@gsb.columbia.edu

**Aravind Srinivasan**
Department of Computer Science
University of Maryland
College Park, MD 20742
asriniv1@umd.edu

## Abstract

Matching is one of the most fundamental and broadly applicable problems across many domains. In these diverse real-world applications, there is often a degree of uncertainty in the input which has led to the study of stochastic matching models. Here, each edge in the graph has a known, independent probability of existing derived from some prediction. Algorithms must probe edges to determine existence and match them irrevocably if they exist. Further, each vertex may have a patience constraint denoting how many of its neighboring edges can be probed. We present new ordered contention resolution schemes yielding improved approximation guarantees for some of the foundational problems studied in this area. For stochastic matching with patience constraints in general graphs, we provide a $0.382$-approximate algorithm, significantly improving over the previous best $0.31$-approximation of Baveja et al. (2018). When the vertices do not have patience constraints, we describe a $0.432$-approximate random order probing algorithm with several corollaries such as an improved guarantee for the Prophet Secretary problem under Edge Arrivals. Finally, for the special case of bipartite graphs with unit patience constraints on one of the partitions, we show a $0.632$-approximate algorithm that improves on the recent $1/3$-guarantee of Hikima et al. (2021).

## 1 Introduction

The offline stochastic matching problem is about finding a maximum matching on a weighted graph. However, each edge $e$ is *active* independently according to a known probability $p_e$, and only active edges can be matched. The set of active edges is initially unknown. An edge whose endpoints are unmatched can be *probed* to determine whether it is active, and if so, it is irrevocably inserted into the matching. The objective is to sequentially probe the edges in a way to maximize the expected weighted matching at the end.

Matching problems arise in numerous deployed AI systems, especially those dealing with allocation and scheduling. See, for example, the works of Ahmadi et al. (2020); Ahmed et al. (2017); Brubach et al. (2020, 2021b); Baveja et al. (2018) for applications to advertising, e-commerce, organ exchange, online dating, peer review, school matching, and hiring; and Hikima et al. (2021); Nanda et al. (2020); Xu et al. (2019) for applications to ride sharing, crowdsourcing (worker-task assignment), and

recommendation systems. The work of Esfandiari et al. (2016); Antoniadis et al. (2020) gives further discussion on applications to e-commerce and internet advertising, and illustrates the importance of designing algorithms with good theoretical guarantees in deployed AI systems.

Stochastic edges play a major role in properly capturing many of these applications when we account for hidden information about whether a proposed match will be accepted. Is a kidney donor a good match? Will a worker accept an offered task? Does a user want to click on this ad? The uncertainty inherent in the presence or absence of various edges naturally leads to the need for stochastic optimization. Achieving good results when the edge probabilities are known demonstrates the value of learning distributions in AI.

We contribute improved algorithms for a range of fundamental problems, as well as useful-special-case problems, in the realm of stochastic matching. Although our results are phrased for the offline setting, some of our algorithms also translate to the *random-order* online setting in which the edges arrive in random order. This online arrival model is well-motivated in these AI settings due to the inability to control the order in which the agents arrive.

Below, we further describe three typical features that may be present in stochastic matching problems.

**Patience constraints.** The offline stochastic matching problem is typically considered with the input including an integral *patience parameter* $t_v$ for each vertex $v$ known as a *patience constraint* or timeout. This adds the constraint that at most $t_v$ of the edges incident to $v$ can be probed, which is motivated by applications in which the vertices represent users who are only willing to view a finite number of potential matches. A vertex $v$ which has been probed $t_v$ times is said to have *timed out*. To capture the absence of patience constraints in some models, we allow patience parameters to be $\infty$.

**Probing orders allowed.** Most generally, we allow algorithms to probe the edges in any order based on the realizations of past probes. Sometimes we impose that the algorithm must make a single pass through the edges, according to a uniform random permutation. This can also represent an "online" setting in which the edges arrive in a uniformly random order, and upon arrival, an edge must either be probed (only possible if both endpoints are unmatched and have remaining patience) or irrevocably discarded. We say that such algorithms are *random-order*.

**Special case of graphs considered.** We introduce a new class of graphs, *bipartite graphs with a unit-patience* side, for which we derive further improved guarantees. Such a graph can be divided into two sides such that: (1) all edges are between vertices on different sides; and (2) for one of the sides, all patiences are 1. This subclass of graphs captures a problem of interest in AI with applications such as crowdsourcing and ride-hailing, described in Hikima et al. (2021).

## 1.1 Summary of Results

To derive computationally-efficient probing algorithms, we address the standard approach of first solving a fractional relaxation, which prescribes a probability $y_e \in [0, 1]$ with which each edge $e$ should be probed. These relaxed values $y_e$ only have to satisfy the matching and patience constraints in expectation, while a probing algorithm must satisfy these constraints with probability (w.p.) 1. However, we can still hope to probe every edge $e$ with probability at least $c \cdot y_e$, for some $c \leq 1$. It is well-known that such an algorithm would then be *c-approximate*, i.e. its expected weight matched would be at least $c$ times that of an optimal probing algorithm.

**Result 1.** A **0.382**-approximate *random-order* probing algorithm for general graphs.

This makes a significant improvement to the line of work on patience-constrained offline stochastic matching for general graphs, which includes the **0.25**-approximate algorithm of Bansal et al. (2012), **0.269**-approximate algorithm of Adamczyk et al. (2015), and **0.31**-approximate algorithm of Baveja et al. (2018). We emphasize that although our algorithm considers the edges in a random order, no better guarantee is known even without imposing this property. Our general guarantee can be parametrized by the patiences in the graph and improves in certain special cases.

**Corollary 1.** A $(1 - e^{-2})/2 \approx$ **0.432**-approximate *random-order* probing algorithm for general graphs without patience constraints.

In the special case where every patience is $\infty$, the guarantee of our random-order probing algorithm improves to 0.432. We note that without patience constraints, the Greedy algorithm which probes edges in decreasing order of weights (ignoring the probabilities $p_e$) guarantees at least 1/2 the offline

maximum weighted matching knowing the set of active edges in advance. However, our algorithm has the benefit of providing a guarantee relative to the relaxed values $y_e$, and moreover only needs to consider the edges in a uniformly random order (instead of needing to choose the order). This means that the special case of our algorithm where patiences are $\infty$ implies new results for *Random-order Contention Resolution Schemes* and *Prophet Secretary under Edge Arrivals*, as we discuss below.

**Corollary 2.** A $(1 - \mathrm{e}^{-2})/2$-selectable Random-order Contention Resolution Scheme for the matching polytope of general graphs.

For $c \leq 1$, the definition of a $c$-selectable Contention Resolution Scheme can be reduced to our goal of probing every edge $e$ with probability at least $c \cdot y_e$, as we explain in Section 2. Previous work in this area has derived a $(1 - \mathrm{e}^{-2})/2$-selectable "offline" Contention Resolution Scheme, which needs to know the set of active edges in advance, for the matching polytope of general graphs (Guruganesh and Lee, 2017). The selectability has been recently improved to be strictly greater than $(1 - \mathrm{e}^{-2})/2$, while also satisfying a monotonicity property (Bruggmann and Zenklusen, 2020). However, both of these results require knowing the set of active edges in advance, while our algorithm observes the activeness of edges in an ordered fashion and must immediately decide whether to insert any active edges into the matching. In fact, our algorithm satisfies the definition of a *Random-order* Contention Resolution Scheme introduced in the papers by Adamczyk and Włodarczyk (2018); Lee and Singla (2018). To this end, our specific setting of the matching polytope is not captured[1] by these papers.

**Corollary 3.** A **0.432**-guarantee for the Prophet Secretary problem under Edge Arrivals, in general graphs.

Contention Resolution Schemes for the matching polytope also imply Prophet Inequalities under the Edge Arrival model introduced in Gravin and Wang (2019). The best-known result here is a **0.337**-selectable *adversarial-order* Contention Resolution Scheme for the matching polytope of general graphs due to Ezra et al. (2020), which implies a 0.337-guarantee for Prophet Inequalities under Edge Arrivals in any order. When the edges arrive in a uniformly random order, this can be called the *Prophet Secretary* problem (Esfandiari et al., 2017) under Edge Arrivals, for which our Random-order Contention Resolution Scheme implies an improved 0.432-guarantee.

**Corollary 4.** A new class of non-adaptive $(1 - 1/\mathrm{e})$-selectable Random-order Contention Resolution Schemes for rank-1 matroids.

In the further special case of a star graph with infinite patiences, the selectability of our probing algorithm improves to $1 - 1/\mathrm{e}$. The constraint that at most one edge in a star graph can be matched corresponds to a rank-1 matroid, for which a $(1 - 1/\mathrm{e})$-selectable Random-order Contention Resolution Scheme is already known (e.g. Lee and Singla, 2018). However, our analysis yields a wide range of new such schemes, which are simpler than existing ones in that they satisfy a *non-adaptiveness* property, which we we elaborate on in Subsection 1.2.

**Result 2.** A $(1 - 1/\mathrm{e}) \approx$ **0.632**-approximate probing algorithm for bipartite graphs with a unit-patience side.

This result improves and generalizes the **1/3**-guarantee of Hikima et al. (2021) which holds for a special case of bipartite graphs with a unit-patience side. Despite the ubiquity of $(1 - 1/\mathrm{e})$-guarantees in online matching, to the best of our understanding, our guarantee requires the novel technical ingredient of a $(1 - 1/\mathrm{e})$-selectable Ordered Contention Resolution Scheme for rank-1 matroids under *negative correlation*. The aforementioned $(1 - 1/\mathrm{e})$-selectability results for rank-1 matroids assume elements to be active independently and do not establish this guarantee, as we discuss in Subsection 1.2. We note that our probing algorithm here must be able to choose the order, though. We note that concurrent and independent work of Borodin et al. (2021) study an online problem which implies the same $(1 - 1/\mathrm{e})$-approximation for the offline problem in the particular case where one side has unit patience and the other side has unlimited patience.

## 1.2 Description of Techniques

**Random-order probing: finding an attenuation function which improves the worst case.** As described before, our probing algorithm for general graphs considers the edges in a uniformly random

---

[1]If the graph is bipartite, then its matching polytope can be captured by the intersection of two matroids; however, in this case the selectability guaranteed by Adamczyk and Włodarczyk (2018) is 1/3, which is worse than our selectability of 0.432.

order. This can be implemented by each edge $e$ drawing an "arrival time" $x_e$ independently and uniformly from [0,1]. Baveja et al. (2018) have previously analyzed a similar algorithm, which probes each incoming edge $e$ according to the fractionally-feasible probability $y_e$ as long as $e$ is safe. They show that every edge $e$ ends up being probed with probability at least $0.31 \cdot y_e$, yielding a 0.31-approximate algorithm. The worst case occurs for an edge $e'$ whose values of $y_{e'}, p_{e'}$ are close to 0, with both endpoints of $e'$ being incident to other edges $e''$ whose values of $y_{e''}, p_{e''}$ are 1.

To improve this worst case, we attach an "attenuation factor" $a(e) \in [0, 1]$ to each edge $e$ such that the probability of an incoming safe edge $e$ being probed is scaled down by a factor of $a(e)$. We make $a(e)$ decreasing in $y_e$ and $p_e$, to dissuade the aforementioned edges $e''$ with large values of $y_{e''}, p_{e''}$ from being probed and blocking edge $e'$. However, given an arbitrary function $a$ defining the attenuation factors, computing the new worst case could be difficult. Therefore, our approach is instead to derive properties on $a$ which cause the *worst case to only involve edges $e$ with $a(e) = 1$*. More specifically, we show that for functions $a$ defined by $a(e) = \tilde{a}(y_e p_e)$ for some univariate function $\tilde{a}$ with $\tilde{a}(0) = 1$, it is possible to design the derivatives of $\tilde{a}$ so that in the worst case, edge $e'$ is only incident to edges $e''$ with $y_{e''} p_{e''} \approx 0$ (which implies that $a(e'') \approx \tilde{a}(0) = 1$). Our final bound is then derived from the same expression as in Baveja et al. (2018), except the worst case ratio has improved to 0.382. The specific attenuation function we compute is inconsequential to this improved ratio—the key is showing the *existence* of an attenuation function which *eliminates* the previous worst case. However, by deriving such necessary properties, we are able to see not only which attenuation functions work, but also which *don't work* (suggesting alternative attenuation functions can't do better than ours). Further, having a *family* of valid attenuation functions allows for a choice of different attenuation functions for different application domains while keeping the same approximation guarantee.

**Using our attenuation for Random-order Contention Resolution Schemes.** As stated in Corollaries 2 and 3, our attenuation analysis in the special case of infinite patiences implies a previously-unknown $(1 - e^{-2})/2$-selectable Random-order Contention Resolution Scheme for the matching polytope of general graphs. We now discuss the further special case in Corollary 4 of star graphs, for which we can contrast our technique with that used in the known Random-order Contention Resolution Schemes for rank-1 matroids. Here, Lee and Singla (2018) show that $(1 - 1/e)$-selectability can be achieved using what we would call the attenuation function $a(e) = \exp(-x_e p_e)$, which depends on the arrival time $x_e$ of each edge $e$. This function is designed (see Ehsani et al., 2018) to yield a closed-form expression for the probability of availability at any particular time $x \in [0, 1]$, which allows them to elegantly compute that the selectability is $1 - 1/e$.

In this special case, our analysis also yields a $(1 - 1/e)$-selectable Random-order Contention Resolution Scheme. However, instead of designing a specific function, our analysis implies a class of functions which sufficiently[2] attenuate large values of $p_e$ to prevent them from blocking smaller values of $p_e$. To elaborate, we show that either of the functions $a(e) = \exp(-\alpha p_e)$ or $a(e) = 1 - \alpha p_e$, with $\alpha \in [1/2, 1]$, ensures that in the worst case, all edges $e$ have $p_e \approx 0$. And in this worst case, $a(e) \approx 1$ for all $e$, from which we can conclude that any edge has an $\approx 1 - 1/e$ chance of being selected. We note that our functions do not depend on the arrival time $x_e$ and can be seen as *non-adaptive* $(1 - 1/e)$-selectable Random-order Contention Resolution Schemes for a rank-1 matroid, which we believe could be applied elsewhere.

**Bipartite graphs with a unit-patience side.** For offline stochastic matching on bipartite graphs, a standard technique (Bansal et al., 2012) is to randomly round the fractionally-feasible values $y_e$ to binary values $Y_e$ using the dependent rounding procedure of Gandhi et al. (2006), which ensures the patience constraints on both sides to be satisfied w.p. 1. Under our additional assumption that one of the sides $V_1$ has unit-patience, the vertices in $V_1$ must have rounded degree at most 1, resulting in the rounded graph being a disjoint collection of stars. One could then separately handle the edges in each star using a Random-order Contention Resolution Scheme for rank-1 matroids, since its edges $e$ will be disjoint from other stars and active independently w.p. $p_e$.

---

[2]This intuition can be illustrated as follows. If there is no attenuation, then the worst case involves two edges $e', e''$ with probabilities $p_{e'} = 0, p_{e''} = 1$, which when shown in a random order implies that $e''$ will block $e'$ w.p. 1/2, whereas $e'$ will not block $e''$. The goal of "attenuation" is to scale down the probability of selecting $e''$, to increase the probability of selecting $e'$.

However, this does not lead to $(1 - 1/e)$-selectability. To elaborate, for any vertex $v \notin V_1$, let $\delta(v)$ denote the set of edges incident to $v$. Fractional feasibility of the $y_e$ values ensures

$$\sum_{e \in \delta(v)} p_e y_e \leq 1, \tag{1}$$

but the rounded star graph formed by edges $\{e \in \delta(v) : Y_e = 1\}$ could have $\sum_{e \in \delta(v)} p_e Y_e > 1$ whenever[3] some particular edge $e'$ has $Y_{e'} = 1$, making it difficult to ensure that edge $e'$ gets selected with sufficient probability when $Y_{e'}$ is rounded up.

**Reinterpretation as contention resolution under negative correlation.** To resolve this issue, we instead imagine each edge $e \in \delta(v)$ as being active with probability $z_e := p_e y_e$, which satisfies $\sum_{e \in \delta(v)} z_e \leq 1$, due to (1). The active edges in $\delta(v)$ are correlated in a way such that they cannot conflict with active edges in other stars; however, due to this correlation, any of the aforementioned Random-order Contention Resolution Schemes which are agnostic to the correlation will only be $1/2$-selectable, as we will show in Section 4.

Despite this apparent lack of a correlation-agnostic $(1 - 1/e)$-selectable Contention Resolution Scheme, what we do show is that the optimal online algorithm, which trivially sorts the edges $e \in \delta(v)$ in decreasing order of weights $w_e$, obtains in expectation at least $1 - 1/e$ times the fractional value $\sum_{e \in \delta(v)} w_e z_e$. This is only possible due to the following *negative correlation* property enjoyed by the rounding procedure of Gandhi et al. (2006), with $Z_e \in \{0, 1\}$ denoting the activeness of an edge $e$:

$$\Pr\left[\bigcap_{e \in S} (Z_e = b)\right] \leq \prod_{e \in S} \Pr[Z_e = b] \qquad \forall S \subseteq \delta(v), b \in \{0, 1\}. \tag{2}$$

Our analysis applies this negative correlation property with $b = 0$ to show that the expected *overall* weight obtained by the online algorithm is *minimized when the $Z_e$'s are independent*, despite the fact that for a *particular* edge $e'$, negative correlation among other edges in $\delta(v)$ *could make it more likely that $e'$ is blocked* than in the independent case. Through the equivalence derived in Lee and Singla (2018), our analysis also implies the existence of a $(1 - 1/e)$-selectable Ordered Contention Resolution Scheme for rank-1 matroids under negative correlation, assuming the order can be chosen. We leave it as an open question whether a $(1 - 1/e)$-selectable *Random-order*[4] Contention Resolution Scheme is possible under property (2).

### 1.3 Roadmap

We begin in Section 2 with some background. Then, in Section 3, we present a random-order algorithm for general graphs, achieving our $0.382$-approximation for stochastic matching. We then analyze the same algorithm in the case of infinite patience, where the algorithm yields a $(1 - e^{-2})/2$-approximation (and thus a $(1 - e^{-2})/2$-selectable Random-order Contention Resolution Scheme for the matching polytope); and the case of a star graph, where the algorithm achieves an approximation guarantee of $1 - 1/e$ (and thus gives a $(1 - 1/e)$-selectable Random-order Contention Resolution scheme for rank-1 matroids). Finally, in Section 4, we present an algorithm for stochastic matching on bipartite graphs with a unit patience side, which achieves an approximation guarantee of $1 - 1/e$. All omitted proofs can be found in the full version of this paper (Brubach et al., 2021a).

## 2 Notation and Preliminaries

The weighted stochastic graph is denoted by $G = (V, E)$, with the weight and probability of being active being denoted by $w_e$ and $p_e$, respectively, for each edge $e \in E$. The patience parameter is

---

[3]As a concrete example, let $v$ have patience $t_v = 2$, and be incident to three edges with $p_1 = 1, y_1 = \varepsilon$; $p_2 = 1 - \varepsilon, y_2 = 1$; and $p_3 = 0, y_3 = 1 - \varepsilon$, which are fractionally feasible in that $\sum_{e \in \delta(v)} p_e y_e \leq 1$ and $\sum_{e \in \delta(v)} y_e \leq 2$. The rounding must be such that whenever $Y_1 = 1$, we also have $Y_2 = 1$, which results in $\sum_{e \in \delta(v)} p_e Y_e = 2 - \varepsilon$.

[4]A $1/2$-selectable Random-order Contention Resolution Scheme for star graphs under negative correlation is implied by the "uniform black box" in Brubach et al. (2020). This black box has also been extended in some cases by Fata et al. (2019).

denoted by $t_v$ for each vertex $v \in V$. Given any problem instance defined by these values, finding the optimal probing algorithm is computationally challenging (Bansal et al., 2012). For $c \leq 1$, a probing algorithm is said to be *c-approximate* if its expected weight matched is at least $c$ times that of the optimal probing algorithm, for any problem instance. The following LP relaxation is commonly used to derive computationally efficient probing algorithms.

$$\text{LP} := \max \sum_{e \in E} w_e z_e \tag{3}$$

$$\text{subject to} \sum_{e \in \delta(v)} z_e \leq 1 \qquad \forall v \in V \tag{3a}$$

$$\sum_{e \in \delta(v)} y_e \leq t_v \qquad \forall v \in V \tag{3b}$$

$$0 \leq y_e \leq 1 \qquad \forall e \in E \tag{3c}$$

$$z_e = y_e p_e \qquad \forall e \in E \tag{3d}$$

Note that we have let $\delta(v)$ denote the set of edges incident to a vertex $v$. The variable $y_e \in [0,1]$ corresponds to the probability of probing edge $e$. The variable $z_e$ is then the probability that edge $e$ is included in the matching (that is, it is both active and probed). Constraint (3a) for a vertex $v \in V$ is the *matching constraint*: it is satisfied when $v$ is matched to at most one of its neighbors in expectation. Constraint (3b) for a vertex $v \in V$ is the *patience constraint*: it is satisfied when at most $t_v$ edges incident to $v$ are probed in expectation.

**Lemma 1** (Bansal et al. (2012)). For any problem instance, the optimal objective value LP is an upper bound on the expected weight matched by any optimal probing algorithm.

Due to Lemma 1, for an algorithm to be $c$-approximate, it suffices to show that its expected weight matched is at least $c \cdot \text{LP}$. All of our algorithms will be based on taking an optimal LP solution given by $(y_e)_{e \in E}$, and randomizing in a way to probe every edge $e$ with probability at least $c \cdot y_e$, which suffices for matching expected weight at least $c \cdot \text{LP}$. We note that the gap between the LP and the optimal probing algorithm can be large: Brubach et al. (2021b) showed that for some graphs, the ratio between the maximum-weight matching and the LP objective value can be as large as $0.544$.

**Definition 1** (Ordered Contention Resolution Scheme Problem). A graph $G = (V, E)$ and a vector $(\tilde{z}_e)_{e \in E}$ lying in its *matching polytope* (i.e. satisfying $\sum_{e \in \delta(v)} \tilde{z}_e \leq 1$ for all $v$) is given. Each edge $e \in E$ has an "activeness" in $\{0, 1\}$ whose state is initially unknown other than that it equals 1 w.p. $\tilde{z}_e$. The activeness of edges is observed sequentially, and if an edge is both active and eligible to be matched (i.e. not incident to any edges already matched), then it can be either immediately matched or irrevocably discarded. A (randomized) algorithm that guarantees every edge $e \in E$ of being matched with ex ante probability at least $c \cdot \tilde{z}_e$ is said to be a *c-selectable Ordered Contention Resolution Scheme* for the matching polytope.

A probing algorithm which guarantees every edge $e$ probability at least $c \cdot y_e$ of being probed implies a $c$-selectable Ordered Contention Resolution Scheme. To see this, given an instance to the problem in Definition 1, we can construct an instance of offline stochastic matching with $p_e = \tilde{z}_e$ for all $e$ and $t_v = \infty$ for all $v$, which means that setting $y_e = 1$ for all $e$ is a feasible solution to LP (3). The probing algorithm will indicate whether to probe each edge in a way that guarantees the overall probability of any edge $e$ being probed to be at least $c \cdot y_e = c$. Therefore, if in Definition 1 we "accept an edge when active" whenever the probing algorithm would have probed that edge, this translates to an ex ante guarantee of $c \cdot p_e = c \cdot \tilde{z}_e$ on the probability of any edge $e$ being matched, as desired.

**Definition 2** (Random-order). A probing algorithm is said to be *random-order* if it can be applied in the online setting where the edges arrive in a uniformly random order, and upon arrival, each edge needs to be either immediately probed (if safe) or irrevocably discarded. Analogously, a *Random-order Contention Resolution Scheme* must observe the activeness of edges in a uniformly random order.

# 3 Algorithm and Analysis for Result 1

## 3.1 Description of Algorithm and Attenuation Functions

Our algorithm is based on the algorithm of Baveja et al. (2018), but with an added attenuation factor. This algorithm first solves LP (3) to get a fractional solution $(y_e)_{e \in E}$. Then, the algorithm independently rounds the LP solution by setting, for each edge $e$, $Y_e = 1$ with probability $y_e$ and $Y_e = 0$ with probability $1 - y_e$. Then, the algorithm generates a uniformly random permutation $\pi$ on $E$; the algorithm then proceeds to probe each edge $e \in E$, in the order of the permutation, if $Y_e = 1$ and both endpoints are still safe (i.e., the endpoints are both umatched with remaining patience).

Our algorithm adds an additional attenuation as follows. Let $a \colon E \to [0,1]$ be our attenuation function; then, when we get to an edge $e = \{u, v\}$, we generate a new Bernoulli random variable $A_e$ such that $\mathbb{E}[A_e] = a(e)$. Then, we probe $e$ if

1. $e$ is *safe*

2. $Y_e = 1$, and

3. $A_e = 1$

Pseudocode is given in Algorithm 1. Recall that we can generate the permutation $\pi$ by first generating, for each $e \in E$, a uniformly random "arrival time" $x_e \in [0,1]$, and ordering the edges in increasing order of arrival time. Slightly abusively, we use $\pi(e) := x_e$ to refer to the arrival time of an edge $e$.

---

**Algorithm 1:** Attenuation-based algorithm for Stochastic Matching

---

**Function** AttenuateMatch($V$, $E$, $\mathbf{p}$):

    Generate a random permutation $\pi$ on $E$

    **for** *each edge $e$ in the order of $\pi$* **do**

        Generate random bit $Y_e = 1$ with probability $y_e$

        Generate random bit $A_e = 1$ with probability $a(e)$

        **if** *e is "safe"* $\land\, Y_e = 1 \land A_e = 1$ **then**

            Probe $e$

---

Our analysis allows for our $0.382$-approximation to be achieved for many choices of attenuation function. Specifically, our analysis will require a few key properties of our attenuation function, outlined in Definition 3 below.

**Definition 3.** We call an attenuation function $a(e)$ *effective* if all of the following conditions hold:

1. $a(e)$ can be expressed as a function $\tilde{a}(z_e)$ of $z_e$

2. $\tilde{a}(0) = 1$

3. $\ln(1 - xz\tilde{a}(z))$ is a *convex* function of $z \in [0,1]$ for any $x \in (0,1)$

There are many functions which satisfy the conditions of Definition 3. Notice, for instance, that the first two conditions are straightforward: we require only that the attenuation function be a function of $z_e$ and result in no attenuation when $z_e \approx 0$. The final condition is satisfied by many nice classes of functions, with some examples given in Definition 4.

**Definition 4.** Define the following attenuation functions:

- The *exponential* attenuation function, defined by $\tilde{a}_{\exp}(z) := e^{-\alpha z}$ for any $\alpha \geq 1/2$

- The *linear* attenuation function, defined by $\tilde{a}_{\lin}(z) := 1 - \alpha z$ for any $\alpha \geq 1/2$

It can be easily verified, by taking second derivatives, that these two functions indeed satisfy the third property of Definition 3, and hence are effective.

## 3.2 Analysis of the Attenuation Algorithm

The expected weight of the matching produced by our algorithm is $\mathbb{E}[\mathsf{ALG}] = \sum_{e \in E} w_e p_e \Pr[Y_e = 1] \Pr[A_e = 1] \Pr[e \text{ gets probed} \mid Y_e = 1 \cap A_e = 1] = \sum_{e \in E} w_e p_e y_e a(e) \Pr[e \text{ gets probed} \mid Y_e = 1 \cap A_e = 1]$. We wish to lower bound $\Pr[e \text{ gets probed} \mid Y_e = 1 \cap A_e = 1]$. We will then establish that for an effective attenuation function, $a(e) \cdot \Pr[e \text{ gets probed} \mid Y_e = 1 \cap A_e = 1] \geq 0.382$. This then implies our 0.382-selectability result for the matching polytope, and establishes the 0.382-approximation for stochastic matching on general graphs.

Fix an edge $e = \{u, v\}$ and consider the endpoint $u$ (the endpoint $v$ can be treated symmetrically).Let $x \in [0, 1]$; we will condition on $\pi(e) = x$. Let $E(u) := \delta(u) \setminus \{e\}$ be the set of $u$'s neighbors *excluding* $e$. If probing an edge $f \in E(u)$ results in either a successful match or $u$ timing out, then we say that edge $f$ *blocks* $e$. We denote by $\overline{B_u}$ the event that $e$ is *not* blocked by any edge $f \in E(u)$.

We wish to analyze the local structure (around edge $e$) that produces the worst bound on $\Pr[\overline{B_u}]$. We now present a sequence of lemmas that lead to the identification of the worst case $e$ for any effective attenuation function.

**Lemma 2.** For effective attenuation functions, all edges $f \in E(u)$ in the worst case have $p_f \in \{0, 1\}$.

As brief intuition for Lemma 2, we note that any edge $f$ with $p_f \in (0, 1)$ can be replaced with two edges $f_0$ and $f_1$ with probabilities of being active of 0 and 1 respectively. We can assign $y_{f_0}$ and $y_{f_1}$ in such a way that all LP constraints are still satisfied, and can show that the probability of an edge blocking $u$ is only increased by doing so.

Equipped with Lemma 2, let $E_0(u) = \{f \in E(u) \mid p_f = 0\}$ and $E_1(u) = \{f \in E(u) \mid p_f = 1\}$. Using the fact that all edges have integer probabilities (either 0 or 1), we can lower bound the probability of $u$ not being blocked by the following quantity:

$$\mathcal{S}_u := \Pr[T_u \leq t_u - 1] \prod_{f \in E_1(u)} (1 - x y_f a(f)) \tag{4}$$

where $T_u = \sum_{f \in E_0(u)} T_f$, and $T_f$ is a Bernoulli random variable with $\Pr[T_f = 1] = x y_f a(f)$. It is easy to see that $\Pr[\overline{B_u}] \geq \mathcal{S}_u$, because: (1) $\Pr[T_u \leq t_u - 1]$ represents the probability of $u$ not timing out before considering edge $e$; (2) the product represents the probability of $u$ not being matched.

The third property of effective attenuation functions (the convexity of $\ln(1 - x z \tilde{a}(z))$) allows us to analyze the worst case behavior of edges in $E_1(u)$ and further simplify our bound. This is stated as Lemma 3.

**Lemma 3.** If $a(f)$ is an *effective* attenuation function, then

$$\mathcal{S}_u \geq \Pr[T_u \leq t_u - 1] \mathrm{e}^{-x(1 - z_e)} \tag{5}$$

We can then state and prove our final results by further analyzing the worst-case behavior of edges in $E_0(u)$.

**Theorem 1** (corresponds to Result 1 from Subsection 1.1). For any graph and any feasible solution $(y_e)_{e \in E}$ to the LP (3), Algorithm 1 using an effective attenuation function considers the edges in a random order and probes every edge $e = \{u, v\} \in E$ with probability at least

$$\left( \tilde{a}(z_e) \int_{x=0}^{1} \mathrm{e}^{-2x(1 - z_e)} g_{t_u}(x) g_{t_v}(x) dx \right) y_e \tag{6}$$

where $g_t(x) := \Pr[T \leq t - 1]$ for $T \sim \mathrm{Pois}(x(t - 1))$. For the effective attenuation functions of Definition 4, the value in parentheses in (6) is minimized when $z_e = 0$. This is then further lower bounded by $\approx 0.382$ when $t_u = t_v = 2$, so the algorithm is a 0.382-approximation.

**Corollary 5** (corresponds to Corollaries 1 to 3 from Subsection 1.1). In the case where all patiences are $\infty$, since $g_\infty(x) = 1$ for all $x \in (0, 1)$, for any feasible solution $(y_e)_{e \in E}$ to LP (3): Algorithm 1, using one of the attenuation functions from Definition 4, considers the edges in a random order and probes every edge $e = \{u, v\}$ with probability at least

$$\left( \int_{x=0}^{1} e^{-2x} dx \right) y_e = \frac{1}{2} \left( 1 - \mathrm{e}^{-2} \right) y_e$$

which is $\approx 0.432 y_e$. In this case, the algorithm is a 0.432-approximation and a 0.432-selectable Random-order Contention Resolution Scheme for the matching polytope.

**Proposition 1** (corresponds to Corollary 4 from Section 1). For any star graph with infinite patiences and any feasible solution $(y_e)_{e \in E}$ to the LP (3), Algorithm 1, using one of the attenuation functions in Definition 4, considers the edges in a random order and probes every edge $e = \{u, v\} \in E$ with probability at least

$$\left( \int_{x=0}^1 \mathsf{e}^{-x} dx \right) y_e = \left( 1 - \frac{1}{\mathsf{e}} \right) y_e.$$ (7)

This yields a $(1 - 1/e)$-selectable Random-order Contention Resolution Scheme for rank-1 matroids which does not adapt to the time of arrival of each element.

## 4   Algorithm and Analysis for Result 2

Let $G = (V, E)$ be a bipartite graph with bipartition $V = V_1 \cup V_2$. Assume $t_u = 1$ for all $u \in V_1$.

**Description of algorithm.** We first solve the standard LP (3) to obtain an optimal solution $(y_e)_{e \in E}$ satisfying $\sum_{e \in \delta(v)} y_e p_e \leq 1$ and $\sum_{e \in \delta(v)} y_e \leq t_v$ for all $v \in V$. Then, we run the rounding procedure of Gandhi et al. (2006) on $y_e$ to get an integral solution $Y_e \in \{0, 1\}$. This guarantees that for each vertex $u \in V_1$ that $\sum_{e \in \delta(u)} Y_e \leq 1$. Thus, at most one vertex $e \in \delta(u)$ will be rounded to $Y_e = 1$ for every $u \in V_1$. Thus, in the rounded graph $\hat{G} := (V, \hat{E})$, where $\hat{E} := \{e \in E : Y_e = 1\}$, each vertex $v \in V_2$ is the center of a star graph. For each vertex $v \in V_2$, we probe the edges $e \in \delta(v)$ in decreasing order of weight, for each $e$ with $Y_e = 1$.

**Analysis of algorithm.** The expected value achieved by this strategy is

$$\mathbb{E}[\mathsf{ALG}] := \mathbb{E}\left[ \sum_{v \in V_2} W(v) \right] = \sum_{v \in V_2} \mathbb{E}[W(v)]$$

where $W(v)$ denotes the weight of the edge matched by the algorithm (if any) for a vertex $v$. In our analysis, we consider each vertex $v \in V_2$ separately, since in the rounded graph, it is the center of a star graph that is disconnected from any other vertices of $V_2$. We first utilize the negative correlation property of our dependent rounding technique (Gandhi et al., 2006) to establish the following lemma.

**Lemma 4.** Consider a fixed vertex $v \in V_2$. Label the edges of $\delta(v)$ from 1 to $k := |\delta(v)|$ such that $w_1 \geq w_2 \geq \cdots \geq w_k$. Then:

$$\mathbb{E}[W(v)] \geq \sum_{i=1}^k w_i z_i \prod_{j=1}^{i-1} (1 - z_j)$$

For $v \in V_2$, let $\mathsf{OPT}(v) := \sum_{e \in \delta(v)} w_e z_e$. Using Lemma 4, we are then able to derive the following, from which our main result immediately follows.

**Lemma 5.** For any vertex $v \in V_2$, we have

$$\mathbb{E}[W(v)] \geq \left( 1 - \frac{1}{\mathsf{e}} \right) \mathsf{OPT}(v)$$

**Theorem 2** (corresponds to Result 2 from Subsection 1.1). For any bipartite graph with a unit-patience side, let $(y_e)_{e \in E}$ denote an optimal solution to the LP (3). Then our algorithm above matches expected weight at least $(1 - 1/\mathsf{e}) \sum_{v \in V_2} \sum_{e \in \delta(v)} w_e p_e y_e$, yielding a $(1 - 1/\mathsf{e})$-approximation.

We note that the problem of stochastic matching on bipartite graphs with a unit patience side is a special case of particular interest, as it captures "Problem A" in Hikima et al. (2021). As shown in Lemma A of Hikima et al. (2021), an $\alpha$-approximation for Problem A implies an $\alpha$-approximation for the *Integrated Stochastic Problem for Control Variables and Bipartite Matching* (ISPCB). This is discussed further at the end of this section.

**Why we cannot use an existing Contention Resolution Scheme.** Lemmas 4 and 5 show that the *total* expected weight collected from edges in $\delta(v)$ is at least $(1 - 1/\mathsf{e}) \cdot \mathsf{OPT}(v)$. We now explain why it is not possible to use a correlation-agnostic Contention Resolution Scheme to match *every* edge $e \in \delta(v)$ with probability at least $(1 - 1/\mathsf{e})$. We consider the following example, in which every edge $e$ has the same value of $z_e = p_e y_e$. Therefore, a correlation-agnostic Contention Resolution

Scheme would treat the edges symmetrically, doing no better than a strategy which considers the edges in a random order until an edge $e$ is matched (which requires both $Y_e$ to be rounded to 1 and for edge $e$ to exist). However, due to the first-stage dependent rounding for $Y_e$, the probabilities of edges being matched end up being negatively correlated (as defined in (2) in the introduction). This negative correlation among the neighbors of a particular edge "0" can increase the probability of edge 0 being blocked to 1/2 (something not possible under independence), as we now demonstrate.

**Example 1.** Consider a star graph with $T + 1$ edges, whose central vertex has patience 2. Take the fractionally-feasible solution $y_0 = 1$, $p_0 = 1/(T+1)$, and $y_1 = \cdots = y_T = 1/T$, $p_1 = \cdots = p_T = T/(T+1)$. Any rounding procedure which satisfies the patience w.p. 1 and preserves the marginal probabilities will set $Y_0 = Y_i = 1$, $Y_{i'} = 0$ for all other $i'$, with $i$ drawn uniformly from $\{1, \ldots, T\}$. This implies that w.p. $1 - 1/(T+1)$, one of the edges $1, \ldots, T$ will match upon being uncountered. Any correlation-agnostic procedure would treat all edges symmetrically, since they all have the same value of $z_i = p_i y_i = 1/(T+1)$. Consequently, edge 0 will have a $(1 - 1/(T+1))/2$ probability of being blocked, whenever it is considered later than the aforementioned edge which matches upon being encountered. Therefore, the correlation-agnostic procedure cannot be better than 1/2-selectable as $T \to \infty$.

### 4.1 Improvement to Approximation Ratio of Hikima et al. (2021)

The work of Hikima et al. (2021) introduces and studies a problem they call *Integrated Stochastic Problem for Control variables and Bipartite Matching* (ISPCB). This problem is a two-stage bipartite matching problem where the algorithm is given a bipartite graph $G = (V_1 \cup V_2, E)$ and must set a control variable $x_u$ for each $u \in V_1$. Then, each vertex $u \in V_1$ *leaves* the graph with probability $p_u(x_u)$ (where $p_u(x_u)$ is some known probability which depends on $x_u$), and then the algorithm computes a maximum-weight matching on the resulting graph.

Hikima et al. (2021) prove an approximation guarantee for ISPCB by first studying a problem they denote Problem A. Problem A can be seen as the problem of stochastic bipartite matching with a unit-patience side which we study here: For each edge $(u, v) \in E$, $p_{uv} = p_u(x_u)$, $t_u = 1$ for each $u \in V_1$, and $t_v = \infty$ for each $v \in V_2$. The proof of Theorem 1 in Hikima et al. (2021) shows that an $\alpha$-approximation of Problem A implies an $\alpha$-approximation for ISPCB. The 1/3-approximation then follows from the 1/3-approximation for stochastic bipartite matching in Bansal et al. (2012). Our Result 2 captures Problem A, which thus implies a $(1 - 1/e)$-approximation for ISPCB, improving on the previous 1/3-approximation of Hikima et al. (2021)

## Acknowledgments and Disclosure of Funding

Nathaniel Grammel and Aravind Srinivasan were supported in part by NSF award CCF-1749864, and by research awards from Amazon and Google.

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
