# OpenReview forum: "Improved Guarantees for Offline Stochastic Matching via new Ordered Contention Resolution Schemes"
_NeurIPS.cc/2021/Conference — NeurIPS 2021 Poster_

### Official Review · Reviewer_Eoae · 2021-07-05

**Rating:** 6
**Confidence:** 4

**Summary:**

They give an improved 0.38 approximation algorithm for a basic stochastic matching problem. They also get improvements for a few other variants. One notable result is improving a recent 0.33 approximation in [Hikima et al] to 0.63.

**Limitations And Societal Impact:**

yes

**Main Review:**

This paper obtains improved approximation algorithms for a number of stochastic matching problems. These problems are motivated from many applications in AI, such as ride hailing, online ads and crowdsourcing.

The most basic problem is the following. There is a weighted graph where each edge e is active with independent probability p_e. The set of active edges is initially unknown. An algorithm can “probe” edges sequentially: if an edge is probed and active then it must be included in the matching. There is also a “patience” constraint that at most t_v edges at node v can be probed. The goal is to maximize the expected weight of the matching. The paper gives a 0.38 approximation for this problem, improving on a 0.31 result (and many previous ones). The main idea is in identifying the correct “attenuation probabilities” in the previous algorithm [Baveja et al.]. Both the algorithm and analysis rely heavily on [Baveja et al.]. They also use the idea from [Adamczyk et al.] for damping the LP probabilities. However, they view this process more abstractly, by searching for an “attenuation function” with certain properties. This seems like a cleaner approach than ad-hoc choices of the damping factor. For a particular attenuation function, they show (numerically) that the approximation is 0.38. Is this the “best” attenuation function, or can one get a better bound with a different function?

An advantage of their approach (and some previous papers) is that the probing order is just random and not chosen in a careful manner. So the results also hold in the “random order” online setting. They also obtain some improvements for problems in this setting.

For bipartite graphs with unit-patience on one side, they get a much better 0.63 approximation. This special case was recently studied in [Hikima et al.] where a 0.33 approximation was obtained. The authors mention that their improved result also implies an improvement in the final result of [Hikima et al.]. This seems like a significant improvement for that problem- but, there are no details given.
---
I'm satisfied that the authors' result improves on the main result in [Hikima et al.]. It will be good to add full details in a later version.

**Time Spent Reviewing:**

6

---

> ### Author Response · Authors · 2021-08-10
> **Thank you for your comments**
>
> Thank you for taking the time to read our paper and write a detailed review. Regarding whether better attenuation functions are possible, our broad set of properties produces the best-possible attenuation functions for the analytical techniques we employ.  That is, our analysis identifies a worst-case graph structure (for the neighborhood of a single edge) which suggests that our attenuation functions are optimal given our analysis.  It is plausible to us that ideas such as analyzing randomized attenuation functions could lead to improved guarantees, which would make the “worst case” more mild; however, identifying such a worst case for a randomized attenuation function appears challenging, and hence we leave it outside the scope of our work.
>
> Regarding the improvement to the result of Hikima et al., this follows from the fact that our (1-1/e)-approximation improves the approximation of “Problem A” used in the proof of Theorem 1 in that paper. Our new proof achieving the (1-1/e)-approximation then replaces Lemma A of Hikima et al., improving the approximation ratio for Problem A from ⅓ to 1-1/e. This improved ratio carries through to the rest of their analysis, so that the ⅓ bound in Theorem 1 becomes 1-1/e, and so the solution to the problem they label (PA) is an (1-1/e)-approximation for their main problem (ISPCB). We apologize for the lack of justification and would be happy to add this explanation to the paper.

---

### Official Review · Reviewer_2oiH · 2021-07-07

**Rating:** 7
**Confidence:** 3

**Summary:**

The submission focuses on the following “stochastic matching” 1-player game: Nature samples a (hidden) graph, where each edge appears independently with probability p_e. The Algorithm can sequentially probe candidate edges. If an edge belongs to the graph, its vertices are matched and can not participate in later matches. The goal is to maximize the total weight of matched edges. In one of the variants, each vertex also has a (known-to-the-algorithm) patience which upper bounds the number of allowed attempted matches.

There is also a second result which generalizes ordered CRS from independent r.v. to negatively correlated.


**Main Review:**

Previous work proposed the natural approach of solving a fractional + in-expectation LP relaxation, and then sequentially probing the edges (in random order) with probabilities obtained from the LP solution. The goal of the analysis is to bound the minimum probability that any edge is probed, compared to its probability in the LP solution (which is an upper bound on the integral but offline optimum).

The authors of the new submission notice that this solution unfairly discriminates low-probability edges: out of k edges w.p. 1/k, we only have probability 1-1/e of picking one of them. In contrast a single edge with probability 1 (which appears identical to the LP) will be picked w.p. 1. This creates a bottleneck in the analysis when low and high probability edges are competing for matching the same vertex. To overcome this issue, the authors propose to weigh down high-probability edges, yielding improved constant factors in the competitive analysis.



I think that this work will be quite interesting, but to a small crowd of specialized experts.


**Time Spent Reviewing:**

2

---

> ### Author Response · Authors · 2021-08-10
> **Thank you**
>
> Thank you for taking the time to read our paper and write a detailed review. We would like to briefly highlight that our paper, despite being largely theoretical, is relevant to several aspects of AI. In particular, our guarantees, which are possible for stochastic matching when the edge probabilities are known, illustrates the value of learning distributions, a cornerstone of modern AI. Additionally, online matching problems are common in deployed AI in a variety of application areas related to our work, such as kidney exchange, online dating, ad allocation, ride sharing, and worker-task crowdsourcing (the last two of which are studied in Hikima et al., 2021). Finally, the importance of worst-case bounds for online matching problems is discussed in several recent NeurIPS papers, with examples being “Bi-Objective Online Matching and Submodular Allocations” (2016) and “Secretary and Online Matching Problems with Machine Learned Advice” (2020).  We acknowledge that our section motivating the problems in AI is a bit brief, and are happy to add more details and cite further papers, including the ones above, in a revised version.

---

### Official Review · Reviewer_1hJb · 2021-07-15

**Rating:** 8
**Confidence:** 4

**Summary:**

The paper considers a version of the stochastic matching problem. We have a graph with known weights on the edges but where each edge is present with probability p_e (also known). Sequentially, a decision-maker has to probe edges. When DM probes an edge, she can see whether the edge was present or not, and in the former case, she must include it in the matching. The main result in the paper is 0.382 approximation-algorithm for this problem.

**Limitations And Societal Impact:**

Do not think it applies here.

**Main Review:**


The main result is quite interesting. It improves on a sequence of earlier work in the area and, quite notably, the 0.382 algorithm is obtained by probing edges in random order!

Several variants and improved factors are discussed. A nice one (obtained as a corollary of the main result) is a best-known-to-date guarantee (of 0.432) for what the authors call prophet secretary under edge arrivals. Here the weights of the edges of a graph are random variables with known distribution and come in random order. A DM has to select edges online to form a matching. The goal is to maximize the weight of the matching, and the benchmark is the expected weight of a max weight matching. There are also many other results. These are a nice complement but somewhat less interesting to me.

One issue that is very interesting (and probably would be nice to have some more discussion in the paper) refers to the differences in the order of arrival of the edges. For the 0.382 algorithm, the authors use random order, but it turns out that nothing better is known even if the algorithm could pick the probing order. On the contrary, for the 0.432 algorithm, again the authors propose to use random order. But this does improve upon the best know algorithm with adversarial order. Is there some sort of general statement connecting these three possible orderings (chosen by ALG, random, adversarial). What happens in the first problem if the order is adversarial? What happens in the second if the order can be chosen by ALG?

Overall I think this is a strong paper. It is, however, purely theoretical. The authors discuss some applications in AI but very en passant and unconvincingly.

**Time Spent Reviewing:**

3

---

> ### Author Response · Authors · 2021-08-10
> **Thank you for your comments**
>
> Thank you for taking the time to read our paper in detail and provide feedback.
>
> Regarding relevance to NeurIPS: We would like to highlight here that our paper, despite being largely theoretical, is relevant to several aspects of AI. In particular, our guarantees, which are possible for stochastic matching when the edge probabilities are known, illustrates the value of learning distributions, a cornerstone of modern AI. Additionally, online matching problems are common in deployed AI in a variety of application areas related to our work, such as kidney exchange, online dating, ad allocation, ride sharing, and worker-task crowdsourcing (the last two of which are studied in Hikima et al., 2021). Finally, the importance of worst-case bounds for online matching problems is discussed in several recent NeurIPS papers, with examples being “Bi-Objective Online Matching and Submodular Allocations” (2016) and “Secretary and Online Matching Problems with Machine Learned Advice” (2020).  We acknowledge that our section motivating the problems in AI is a bit brief, and are happy to add more details and cite further papers, including the ones above, in a revised version.
>
> Regarding the relationship between adversarial, random-, and chosen-order arrivals, certainly adversarial arrivals can do no better than random, which in turn can do no better than chosen-order. For many problems, adversarial arrival order can often present a strong challenge, and is sometimes even intractable. For instance, it is known that no constant competitive ratio is possible for online bipartite matching with edge weights, under adversarial vertex arrivals (see the survey “Online matching and ad allocation” by Mehta, 2013); this justifies our assumption that the edges can be considered in a random or chosen order. Gaps between random and chosen order are not so severe, and ours is not the only problem for which the best chosen-order algorithm is also the best random-order one (see, e.g.,  https://www.sigecom.org/exchanges/volume_17/1/CORREA.pdf, where the best result for the classical prophet-inequality problem when the order can be chosen by the algorithm is achieved by an algorithm that uses a random order and hence also applies to the random-order prophet secretary problem).

---

### Official Review · Reviewer_PbEY · 2021-07-20

**Rating:** 7
**Confidence:** 3

**Summary:**

The paper considers the offline stochastic matching problem: Each edge e of the graph is equipped by a probability p_e (that it exists) and a weight w_e.
The goal is to find a good probing strategy: an edge can be probed if we can add it to the current matching, if probed and it exists it has to be added to the matching. One difficulty is that vertices have patience constraints stipulating how many adjacent edges the strategy can probe.

The main results of the paper are

- An improve approximation guarantee of 0.383 for general graphs; improving upon the previous best 0.31.

- An improved guarantee of 0.432 if vertices have unbounded patience (it is trivial to get 1/2 by querying the edges in decreasing order; however their result applies in the random edge-arrival online model)

- In the special case of bipartite graphs and one side has unit patience, they get an 1-1/e guarantee improving upon a recent 1/3-guarantee.

The results are nice and are obtained by interesting insights.

**Limitations And Societal Impact:**

See the last paragraph in the main review.

**Main Review:**

The paper provides original and interesting results.

The improved approximation guarantee for general graphs heavily builds upon a prior work and adds the idea of "decreasing the probability of very likely edges". This is a quite well-known trick but still leads to the improved guarantees in this setting.  I am not certain why a family of attenuation functions is better than having one that works.

The result for bipartite graphs with unit patience on one side is obtained by a pretty natural negative correlation rounding.

All their results also apply to the random order online setting  except for the last mentioned bipartite case. This leads to interesting contention resolution schemes for general graphs and results for the prophet secretary problem in that setting.

I find the result for rank-1 matroids less interesting. The Lee and Singla result apply to all matroids with the same guarantee and I am not convinced that the presented schemes are much better.

Overall, I find it quite a nice collection of results. A concern is why NeurIPS would be the right venue. The authors should do a better job explaining why this is related to learning (now they simply cite a couple of papers but they don't spend much effort in explaining the relationship). Also a more detailed comment is if there is any known integrality gaps of the studied LP?

**Time Spent Reviewing:**

5

---

> ### Author Response · Authors · 2021-08-10
> **Thank you for your review**
>
> Thank you for taking the time to review our paper in detail.
>
> Regarding relevance to NeurIPS: Our guarantees, which are possible for stochastic matching when the edge probabilities are known, illustrates the value of learning distributions, a cornerstone of modern AI. Additionally, online matching problems are common in deployed AI in a variety of application areas related to our work, such as kidney exchange, online dating, ad allocation, ride sharing, and worker-task crowdsourcing (the last two of which are studied in Hikima et al., 2021). Finally, the importance of worst-case bounds for online matching problems is discussed in several recent NeurIPS papers, with examples being “Bi-Objective Online Matching and Submodular Allocations” (2016) and “Secretary and Online Matching Problems with Machine Learned Advice” (2020).  We acknowledge that our section motivating the problems in AI is a bit brief, and are happy to add more details and cite further papers, including the ones above, in a revised version.
>
> Regarding your question about why we provide a family of attenuation functions: indeed, a single concrete function is sufficient to get our theoretical approximation results, and we give a few examples in Definition 4.  However, we write our proof based on an abstract function that satisfies our “effectiveness” properties in Definition 3 because it helps to highlight the extent and limitations of our analysis. That is, one may be tempted to ask “what other attenuation functions might work, and which ones might even give better results?” Our more-abstract characterization suggests that our analysis will give the same result for many different functions (any function satisfying the given properties will achieve the same bound, using our analysis), thus suggesting that getting an even-better approximation ratio will likely require an attenuation function with even stronger properties along with a new analysis. More practically, the 0.38 factor refers to the worst-case scenario; real-world data may perform much better on average, and the real-world performance may vary with different attenuation functions. This means that different application domains may be better served in practice by different attenuation functions, and that our result can be applied to conclude that even functions tailor-made to a specific application can achieve the same theoretical guarantees provided they satisfy the given properties.
>
> Regarding the (1-1/e)-approximation, we have two such results (one simply states that our 0.38-approximation algorithm achieves an improved approximation in the case of star graphs with infinite patience, while the other is a new scheme for bipartite graphs). For a discussion on what our new scheme for bipartite graphs (with a unit-patience side) is able to establish beyond Lee and Singla (2018), please see our explanation under “Why we cannot use an existing Contention Resolution Scheme” on page 9.  Certainly the Lee and Singla result is more general in that it applies to all matroids, but to the best of our knowledge, our scheme is needed when negative correlations are present.
>
> Regarding integrality gaps for the fractional LP we use, there is an upper bound in “Follow your star: New frameworks for online stochastic matching with known and unknown patience” (AISTATS, 2021) which shows that the expected size of the offline maximum matching in a general graph can be as small as 0.544 times the objective value of a specific LP.  Therefore, any actual probing algorithm cannot hope to have a guarantee better than 0.544 when analyzed against that LP.  To our knowledge, there are no tighter upper bounds known for a sequential probing algorithm, although that is an interesting question.  We will be sure to add these points to our paper. We also note that gaps for these problems are not exactly “integrality gaps” since fractional solutions representing probing probabilities can be allowed. The aforementioned 0.544 example is called a stochasticity gap.

---

> > ### Comment · Reviewer_PbEY · 2021-09-01
> > **re**
> >
> > Thanks for the informative rebuttal.

---

### Decision · Program_Chairs · 2021-09-27

**Decision:**

Accept (Poster)

**Comment:**

All of the reviewers liked this paper and felt that it should be accepted.